# Simulation Studies Provide Evidence of Aerosol Transmission of SARS-CoV-2 in a Multi-Story Building via Air Supply, Exhaust and Sanitary Pipelines

**DOI:** 10.3390/ijerph19031532

**Published:** 2022-01-29

**Authors:** Zhuona Zhang, Xia Li, Qin Wang, Xiaoning Zhao, Jin Xu, Qinqin Jiang, Sili Jiang, Jiayun Lyu, Shiqiang Liu, Ling Ye, Jun Yuan, Wenru Feng, Jiamin Xie, Qiuling Chen, Haoming Zou, Dongqun Xu

**Affiliations:** 1National Institute of Environmental Health, China Center for Disease Control and Prevention, Beijing 100021, China; zhangzhuona@nieh.chinacdc.cn (Z.Z.); lixia@nieh.chinacdc.cn (X.L.); wangqin@nieh.chinacdc.cn (Q.W.); 2Section of Ecological Environment & Energy Resources, Beijing Institute of Metrology, Beijing 100012, China; zhaoxiaoning@bjjl.cn; 3China National Center for Food Safety Risk Assessment, Beijing 100021, China; xujin@cfsa.net.cn; 4Guangzhou Center for Disease Control and Prevention, Guangzhou 510440, China; jiangqinqin1509@163.com (Q.J.); Sili_jiang@163.com (S.J.); Showst@163.com (J.L.); chemistry20211215@163.com (S.L.); yuanjuncom@163.com (J.Y.); gzcdchwk@163.com (W.F.); 5Guangdong Field Epidemiology Training Program, Guangzhou 511430, China; yeling2680@163.com (L.Y.); gdfetp16xjm@163.com (J.X.); yfepi@163.com (Q.C.); zhmzjcdc@126.com (H.Z.); 6Heyuan Municipal Center for Disease Control and Prevention, Heyuan 517000, China; 7Shunde District Center for Disease Control and Prevention, Foshan 528300, China; 8Yunfu Municipal Center for Disease Control and Prevention, Yunfu 527300, China; 9Zhanjiang Municipal Center for Disease Control and Prevention, Zhanjiang 524037, China

**Keywords:** quarantine hotel, field simulation of SARS-CoV-2, transmission path, air supply, drainage

## Abstract

A cross-layer non-vertical transmission of severe acute respiratory syndrome coronavirus 2 (SARS-CoV-2) occurred in a quarantine hotel in Guangzhou, Guangdong Province, China in June 2021. To explore the cross-layer transmission path and influencing factors of viral aerosol, we set up different scenarios to carry out simulation experiments. The results showed that the air in the polluted room can enter the corridor by opening the door to take food and move out the garbage, then mix with the fresh air taken from the outside as part of the air supply of the central air conditioning system and re-enter into different rooms on the same floor leading to the same-layer transmission. In addition, flushing the toilet after defecation and urination will produce viral aerosol that pollutes rooms on different floors through the exhaust system and the vertical drainage pipe in the bathroom, resulting in cross-layer vertical transmission, also aggravating the transmission in different rooms on the same floor after mixing with the air of the room and entering the corridor to become part of the air supply, and meanwhile, continuing to increase the cross-layer transmission through the vertical drainage pipe. Therefore, the air conditioning and ventilation system of the quarantine hotel should be operated in full fresh air mode and close the return air; the exhaust volume of the bathroom should be greater than the fresh air volume. The exhaust pipe of the bathroom should be independently set and cannot be interconnected or connected in series. The riser of the sewage and drainage pipeline of the bathroom should maintain vertical to exhaust independently and cannot be arbitrarily changed to horizontal pipe assembly.

## 1. Introduction

In June 2021, a cross-layer non-vertical transmission of severe acute respiratory syndrome coronavirus 2 (SARS-CoV-2) occurred in a ten-story quarantine hotel in Guangzhou, Guangdong Province, China. During quarantine, the early quarantined persons on the fifth floor and the late quarantined person in the non-vertically adjacent room on the seventh floor were detected as successively positive for the novel coronal nucleic acid. The index cases and infected case at a different layer had no direct contact before and after entering the hotel and were not the close contacts of the same confirmed case of coronavirus disease 2019 (COVID-19), but the virus gene sequencing results were highly consistent. After investigation, there were no external windows of the rooms they lived in, the air conditioning system of the rooms was a fan coil unit and the central air conditioning air supply system to supply air partly taken from the outdoor and partly from the corridor, the sewage and drainage pipes in bathrooms of the fifth to eighth floor for quarantine did not directly exhaust from the top through the vertical pipes but were gathered to a horizontal pipe at the top of the eighth floor and then exhausted through the roof exhaust fan, which was always closed due to elevator maintenance during the quarantine period.

The WHO pointed out on 30 April 2021: “The virus can also spread in poorly ventilated and/or crowded indoor settings, where people tend to spend longer periods of time. This is because aerosols can remain suspended in the air or travel farther than conversational distance (this is often called long-range aerosol or long-range airborne transmission).” During the outbreak of the COVID-19 epidemic, high-risk populations such as close contacts needed home isolation or centralized isolation for a long time. Usually, the isolation sites were multilayer or high-rise buildings and people were more concentrated, so it was very important to explore virus transmission in the building. Previous viral transmissions in high-rise buildings, including the severe acute respiratory syndrome coronavirus 1 (SARS-CoV-1) outbreak in Amoy Garden in Hong Kong [1,2] and SARS-CoV-2 transmission in a high-rise building in Guangzhou [3] and so on, usually took place in vertically distributed rooms on different floors where index cases and infected cases stayed, and the virus was transmitted through vertical sanitary pipelines in the form of aerosol. In addition, viral aerosol can be transmitted from one building to an adjacent building (commonly known as handshake buildings) under a certain airflow layout that was confirmed in a study about SARS-CoV-2 in Guangzhou [4].

To explore the transmission path and influencing factors of SARS-CoV-2 aerosol in quarantine hotels, we simulated respiration and toilet flushing after defecation and urination of the quarantined person, conducted field simulation experiments, and proposed rectification suggestions for quarantine hotels according to the research results.

## 2. Materials and Methods

The simulation site of this survey was a hotel in Guangzhou, Guangdong Province, China that was used for quarantine of close contacts during the epidemic period. The hotel building showed a rectangular-ambulatory-plane structure. The early quarantined persons on the 5th floor came from the same family in a community and were distributed in rooms 510, 511, 552, which were relatively close. The possibility of infection before occupancy or personnel contact during occupancy was not excluded. The late infected case in room 705 on the 7th floor, which was non-vertically distributed with the three rooms on the 5th floor, had never contacted them but the virus gene sequencing results were highly consistent.

At present, viable novel coronavirus has been detected in both the exhaled air [5] and feces [6,7] of COVID-19 patients. Therefore, we selected two scenarios to simulate respiration in the bedroom and toilet flushing after defecation and urination in the bathroom for field simulation experiments. During the quarantine period, quarantined persons need to stay in a closed indoor space for a long time so as to generate a relatively large amount of aerosol. Meanwhile, the asymptomatic or COVID-19 case in the incubation period also have diarrhea symptoms [8], so toilets will be frequently used during the quarantine period. Based on this, two modes of virus excretion lasted for 4 h and 1.5 h (flushing four times), respectively. The fluorescent polystyrene microspheres consistent with the similar aerodynamic characteristics of the SARS-CoV-2 spike pseudovirus which did not exist in nature were used as the simulants for field simulation experiments. Fluorescent microspheres (0.1–2 μm, Beijing Institute of Metrology, Beijing, China) were prepared in simulated body fluid (SBF). The amounts of simulants were 10^11^ balls per pouring in the scenario of toilet flushing. Collison Nebulizer (BGI, INC, Waltham, MA, USA) was used to simulate respiration of the infected person and generated 10^12^–10^13^ balls per hour. Specific methods were described in previously published papers [4,9,10].

The respiratory simulation experiment lasted for a long time and produce a large amount of aerosol. To prevent the accuracy of the monitoring results from being affected, toilet flushing simulation with a small amount of aerosol after defecation and urination (scenarios 1) was selected first and room 552, basically matching the pattern of room 05, was chosen to simulate defecation and urination to avoid the background pollution in room 705, which would be monitored. The sewage and drainage pipes and air ducts of toilets on different floors were vertically connected. In this experiment, rooms 652 and 752 were monitored and the toilets were flushed simultaneously when pouring simulants and toilet flushing in room 552 to explore the influencing factors of the vertical transmission of the simulants. Considering that the viral gene sequencing results of the early quarantined persons in rooms 510, 511, and 552 and the person in room 705 were all consistent, the three rooms were all selected to simulate respiration to produce viral aerosols (scenarios 2) and rooms 553, 652, and 705 were monitored to investigate the transmission paths and influencing factors of the same layer, vertical cross-layer, and non-vertical cross-layer transmission of simulated aerosols. The specific simulation scenarios are shown in Appendix A. After the beginning of the experiment, the concentration of particles with different sizes in the air was monitored with a particle size spectrometer and PM_10_ measuring instrument every 10 min to determine the change of simulated aerosol with time. PM_10_ samplers (100 L/min) and bioaerosol samplers (100 L/min) were used to collect air aerosol samples at different rooms. The smear samples at different positions were collected at different times to determine the transmission path. The specific sampling method, location, time, and so on are shown in Appendix A. The behavior of opening the door to throw garbage and to take food was simulated during the experiment. The simulation experiment tried to recover the room environment conditions during quarantine: the door of the room was closed, and the air conditioning and fresh air supply system were opened (the room temperature was 24–28 °C and the humidity was 60–70%); the exhaust fan of the bathroom was opened (the flow velocity was 0.1–0.25 m/s).

After the field simulation experiment, the samples were collected and brought to the laboratory. The filter membrane samples were cut in equal proportion. The smeared swab samples were wetted with 75% ethanol and smeared on the clean slides. Liquid samples of different depths in the sampling bottle were absorbed by a pipettor and dropped onto a clean slide for observation. The yellow or green fluorescent particles in different samples were observed by fluorescence microscope (Nikon DS-Ri). The data directly read from the field were analyzed by OriginPro 8 SR3 (OriginLab Company, Northampton, MA, USA).

## 3. Results

### 3.1. Simulated Aerosol Transmission Caused by Toilet Flushing

The sewage and drainage pipes of bathrooms on the fifth to eighth floor inhabited by the quarantined persons gathered to a horizontal pipe at the top of the eighth floor and then exhausted through a roof exhaust fan. Mechanical exhaust systems were adopted for bathrooms where two small exhaust fans were set up above the toilet and the shower room, and the exhaust gas was discharged into the air duct and then discharged from the roof.

In a toilet-flushing simulation experiment after defecation and urination for 1.5 h, fluorescent polystyrene microspheres (0.1–2 μm) were detected in air filter membrane samples and liquid samples collected in rooms 552, 652, and 752, all smeared swab samples collected in room 552 and on the exhaust outlet and sewage pipe riser vent on the roof. The results are shown in Table 1 and Figure 1 and Appendix A. When flushing simultaneously in rooms 552, 652, and 752 after the simulants were poured into the toilet in room 552 at 0 h, no change in the particle concentration occurred in any of the three rooms; when flushing in the three rooms at the same time at 0.5 h, significant peak concentrations were observed at different particle sizes and the background value was restored in about 20 min. The number concentration of particles in different rooms was room 552 > 652 > 752, and fluorescent microspheres were detected in swab samples of the exhaust fans in room 652 and 752. Pouring the simulants to flush again at 1 h, the concentration of particles did not change immediately, and the peak concentration appeared in different rooms at different times (10–30 min). Fluorescent microspheres were detected in swab samples collected on the floor drain, toilet lid and seat ring, and exhaust fan in the bathroom of room 652. Flushing simultaneously at 1.5 h, fluorescent microspheres were detected on the floor drain, toilet lid and seat ring, air supply outlet of the air conditioning, and fresh air system of room 652, and the exhaust fan and floor drain of room 752. Detailed changes in the particle size spectrum are shown in Appendix A.

### 3.2. Aerosol Transmission Induced by Simulated Respiration

Fan coil units and fresh air conditioning systems were in place in the hotel. The fresh air inlets were set on both sides of each floor corridor near the outer window to take the air outside the window. Fresh air was mixed with air from the inside corridor and supplied to the room through two air outlets in the room. The fan coil unit adjusted the room temperature by self-circulating cooling.

The respiratory simulation experiment lasted for four hours; the results are shown in Table 1 and Figure 1 and Appendix A. Fluorescent microspheres were detected in the air filter membrane samples and liquid samples collected in rooms 553, 652, and 705 every two hours, the swab samples collected at different times in rooms 510, 511, and 552, where respiration was simulated, the swab samples collected on the roof exhaust outlet, and the sewage pipe riser vent on the roof. The increasing concentrations of particles with different sizes were monitored in rooms 510, 511, and 552. After the 1 h simulation experiment, fluorescent microspheres were not detected in swab samples of rooms 553, 652, and 705. However, the concentrations of particles with different sizes increased in room 652, did not change in room 705, and was not monitored in room 553 due to the limited field equipment. The increasing particle concentration was also observed under the fresh air inlets on both sides of the fifth corridor and in the outdoor corridor of the room with simulated respiration, and the higher the particle concentration in the room, the higher the corridor. In 2 h, fluorescent microspheres were detected in swab samples collected in room 553, air supply outlets of air conditioning and fresh air system in room 652, and only the air supply outlet of the air conditioning system in room 752. The number concentration of particles in rooms 652 and 705 increased significantly. In 3 h, fluorescent microspheres were detected in swab samples of rooms 553, 652, and 705. The PM_10_ concentration in the room was also monitored, and significant increases were observed in about 40 min in room 553, about 60 min in room 652, and 100 min in room 705. Detailed changes in the particle size spectrum are shown in Appendix A.

## 4. Discussion

### 4.1. Cross-Layer Vertical Transmission Caused by Sewage and Exhaust Pipe of the Bathroom

The results of the simulation experiment showed that the simulated toilet flushing after defecation and urination in room 552 can produce aerosols and spread to the upper bathrooms in 652 and 752. The schematic diagram of vertical transmission is shown in Figure 2. When flushing the toilet, the pressure generated by flushing would cause the fluorescent microspheres to form aerosols [11,12,13], partly into the bathroom and partly into the sewage pipe. The simulated virus aerosol into the bathroom entered the air duct through the exhaust fan. The virus transmission events and field simulation experiments occurred in June, and the outdoor temperature could reach 34 °C. The open indoor air conditioning led to a lower temperature than the outdoors, so the warm air outdoors would flow down into the air duct. Under a certain wind direction, the air in the air duct can pour into other rooms with the vertical distribution. The positive results of smeared swab samples of exhaust fans in three rooms and the roof exhaust outlet after the experiment confirmed that the simulated aerosol can be transmitted through the exhaust duct.

Due to the reconstruction of the sewage and drainage pipes in the fifth to eighth bathroom, the virus aerosol entering the sewage pipe first converged into the horizontal pipe and was then discharged by the exhaust fan at the roof through the riser vents. During quarantine, the exhaust fan of the roof riser has not been turned on; viral aerosol constantly converged in the horizontal pipe and could not be discharged normally. With the change in pressure in the pipe caused by toilet flushing, it would be poured into the sewage and drainage pipe of other floors, and then transmitted to the bathroom through the dry floor drain or washbasin. This hypothesis was confirmed by the positive results of swab samples of the sewage roof riser vents and the floor drain and the toilet seat ring in rooms 652 and 752.

With the increase in simulated aerosols in the air, the concentration of particles will appear at a peak, but it will return to the background in about 20 min, and the peak time of different rooms was different. In addition, the detection time of fluorescent microspheres in different locations of different rooms was also different, indicating that the time of aerosols reaching different floors was not consistent. Diffusion of viral aerosols accumulated in exhaust pipes was related to the direction and velocity of external airflow. The aerosol generated by room 552 spread faster in the exhaust pipe and was transmitted to rooms 652 and 752 in about 0.5 h. The simulated aerosols entering the sewage pipes spread slowly in the pipeline and entered the bathroom of room 652 at the third flush and the bathroom of room 752 at the fourth flush. In addition, with the increase in the amount of simulated aerosol entering the bathroom in room 652, it spread to the bedroom in room 652 and polluted the air conditioning system.

### 4.2. Same-Floor Transmission Caused by Ventilation Systems

The rooms in this hotel were relatively confined because the room had no outer windows and the door was closed during the quarantine period. The viral aerosol generated by simulated respiration gradually increased in the room that can be diffused into the corridor through the door gap or open the door to take food and move out the garbage and other behaviors. The fresh air system mixed the fresh air taken from the outside and the polluted air in the corridor to send to the rooms on the same floor and polluted these rooms. The positive results of swab samples at the fresh air outlet in the rooms on the fifth floor indicated that the fresh air system was polluted and the positive results of air samples and swab samples on the surface in room 553 proved that the room on the same floor was polluted. It can be speculated that the simulated aerosol could also be transmitted to rooms such as room 505. The monitoring and sampling results at different times showed that the simulated aerosol can pollute the fresh air system and be transmitted in the same layer within 1 h of the simulated respiration experiment. The schematic diagram of the transmission in the same layer is shown in Figure 3.

### 4.3. Cross-Layer Non-Vertical Transmission Caused by the Superposition of the Exhaust System in the Bathroom and the Ventilation System in the Room

The airflow carrying viral aerosols in the rooms on the fifth floor continued to flow to the bathroom because the exhaust fan was turned on and entered the exhaust duct. The airflow in the duct was poured into the bathrooms on the sixth and seventh floor under a certain wind direction, causing cross-layer vertical transmission. The positive results of swab samples at the roof exhaust outlet after the simulation experiment further confirmed this conclusion. At the same time, the air in the bathrooms of the sixth and seventh floors was mixed with the air in the bedroom, which polluted the air in the bedroom and entered the corridor. The fresh air system running in mixed mode sent polluted air to different rooms on the same floor, which aggravated the aerosol pollution in the room. The positive results of swab samples on the fresh air supply outlets of rooms 652 and 705 confirmed the pollution of the fresh air system on the sixth and seventh floors. The application of the bathroom exhaust system and the room air supply system caused the cross-layer non-vertical transmission. In addition, the fecal aerosols after toilet flushing would continue to cause cross-layer transmission through sewage pipes. Vertical and non-vertical cross-layer transmission was observed within 2 h in the simulated respiration experiment. The longer the distance of virus aerosol transmission was, the longer the time required.

### 4.4. Influence Factors of Aerosol Transmission

Virus aerosols can spread through the toilet sewage system [13,14,15] and the room ventilation system [16,17,18]. In the SARS-CoV-1 outbreak in Amoy Garden of Hong Kong in 2003, an on-site investigation indicated that the U-shaped water trap of the sanitary fixture played an important role in the vertical transmission of the different floors [1]. In an outbreak of COVID-19 in vertically aligned flats in a high-rise building, Kang et al. used ethane tracer gas in the bathroom for airflow and dispersion tests and clarified the importance of U-shaped water traps and bathroom ventilation [3]. In two housing blocks in Hong Kong, Wang et al. used SF6 tracer gas to conduct toilet simulation experiments and believed that SARS-CoV-2 could be transmitted by long-range aerosol through drainage pipes [19]. Li et al. adopted computational fluid dynamics to model the trajectories of aerosol particles during toilet flushing and observed massive upward transport of virus particles [13]. The above studies provided evidence for virus transmission through sanitary pipes from the aspects of field investigation, experiment, and computer simulation, which were consistent with our experimental results. However, in several experimental studies, tracer gas cannot simulate the agglomeration and resuspension of particles, while SARS-CoV-2 was a particle with a diameter of 100 nm [7] which can aggregate or agglomerate under certain humidity, so the results may have some shortcomings. In our experiment, fluorescent polystyrene microspheres consistent with the similar aerodynamic characteristics of the SARS-CoV-2 spike pseudovirus were used to investigate the effects of the toilet sewage system and exhaust system on the transmission of virus aerosol, which can better simulate the aerosol transmission characteristics of SARS-CoV-2.

Central air conditioning and air supply systems were the transmission routes of infectious diseases such as influenza and SARS-CoV-1 [20,21]. The usage, ventilation rate, and air exchange rate of the air supply system all affected the transmission of the virus. The N and E genes of SARS-CoV-2 were detected by Nissen et al. 2020 in ventilation vents of a COVID-19 ward in Sweden. The central ventilation HEPA filters were found to be positive for both genes in samples from adjacent wards, indicating that the virus can be transported through the central conditioning system for long-distance transmission [22]. Horve et al. also collected SARS-CoV-2 RNA on air handling units in a healthcare setting, which also showed that the virus may spread through the air conditioning system in the room [23]. Our research proved that the use of air conditioning and air supply systems played a key role in virus transmission through field simulation experiments.

## 5. Conclusions

In this study, two different scenarios of simulation experiments of toilet flushing after fecal and urine excretion and respiration were used to explore the path and influencing factors of viral aerosol transmission, including sewage and exhaust pipes, air conditioning, and the air supply system. The results showed that the mixed mode of the fresh air system in the quarantine hotel would mix the polluted air dispersed from the room with the fresh air outside the window and sent it to different rooms on the same floor, resulting in the transmission of simulated aerosols on the same floor. Although the fresh air system had filter devices, the filters that were not often disinfected or replaced not only did not purify the polluted air but also became sources of pollution after running for a certain time. When the novel coronal nucleic acid of quarantined person was detected as positive, caused the infection risk of other persons on the same layer and cross-layer. The investigation and simulation experiments have confirmed that in the case that the fan was not in normal use, the viral aerosol would accumulate in the sewage pipe collected by the horizontal pipe and could not be discharged, which would enter the rooms of different floors through the dry floor drain and the washbasin. In addition, the up-and-down toilet exhaust systems were easy to make the airflow carrying viral aerosols flow into other floors in a specific wind direction, causing vertical cross-layer transmission. The superposition of the toilet sewage and exhaust system and room ventilation system would aggravate the risk of viral aerosol transmission, causing non-vertical cross-layer transmission.

At present, people have a preliminary understanding of SARS-CoV-2 [24,25], but outbreaks still occur in different countries around the world. In epidemic prevention and control, it is very important to find the source of infection that is an index case and cut off the transmission path in time. The confirmation of the index case is closely related to various factors. Even if the sequencing results of the novel coronavirus gene of the infected case are completely consistent with those of the early case, it is necessary to determine a clear transmission path to judge the index case. In addition, nosocomial infection incidents have occurred; other than nonstandard operations of medical staff, some infections were caused by environmental factors. However, the factors in different regions have various kinds of complexity and interfere with each other, so it is difficult to accurately judge by field investigation or computer simulation. In the field simulation experiment, different simulation scenarios can be set according to different influencing factors, and fluorescent polystyrene microspheres consistent with the similar aerodynamic characteristics of the SARS-CoV-2 spike pseudovirus can be used to determine the transmission path of the virus and the main influencing factors, so as to provide the basis for the relevant departments to take measures to achieve precise prevention and control, effectively cutting off the source of infection, controlling the spread of the virus.

## 6. Limitations

This field simulation experiment was a qualitative study. There was a certain deviation between the transmission characteristics of the simulant aerosol and SARS-CoV-2 aerosol as the viable virus cannot be used in the field. The viable virus and its load in aerosols cannot be simulated by fluorescent polystyrene microspheres so the risk of infection cannot be quantitatively predicted; in addition, there was a difference between simulated aerosol amounts and the reality of the confirmed case. Therefore, the field simulation experiment can only provide the transmission path and influencing factors of viral aerosol.

## 7. Sanitation Requirements for Environmental Management in Quarantine Hotels

During quarantine, hotels with windows open to the outside should be chosen to prevent the risk of cross-infection of quarantined persons, and natural ventilation is preferred; windows should be regularly opened for ventilation every day. Hotel managers need to conduct an examination of the hotel’s environmental risks to eliminate the hidden dangers of drainage pipes, the central air conditioning supply system, and bathroom exhaust system before using as a quarantine hotel. This can be achieved as follows: the air conditioning and ventilation system should adopt the full fresh air mode and turn off the return air to avoid using contaminated air in the corridor to mix with fresh air and supply to the room. The exhaust volume of the bathroom should be greater than the fresh air volume. The exhaust pipes of the bathroom need to be set up independently and cannot be interconnected or connected in series. The riser of the sewage and drainage pipeline of the bathroom should be maintained vertical to exhaust independently and cannot be arbitrarily changed to horizontal pipe assembly. The floor drain in the bathroom should be ensured to have a good water seal. The central air conditioning supply system should be cleaned and disinfected, and its filters should be replaced regularly.

## Figures and Tables

**Figure 1 ijerph-19-01532-f001:**
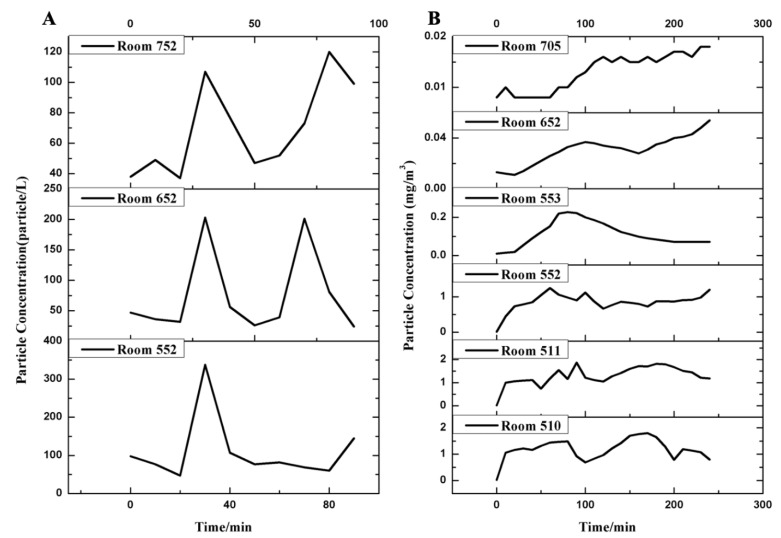
The changes of particle concentration over time at 0.5 μm (scenarios 1), PM_10_ (scenarios 2) in different rooms. (**A**) Scenarios 1; (**B**) scenarios 2.

**Figure 2 ijerph-19-01532-f002:**
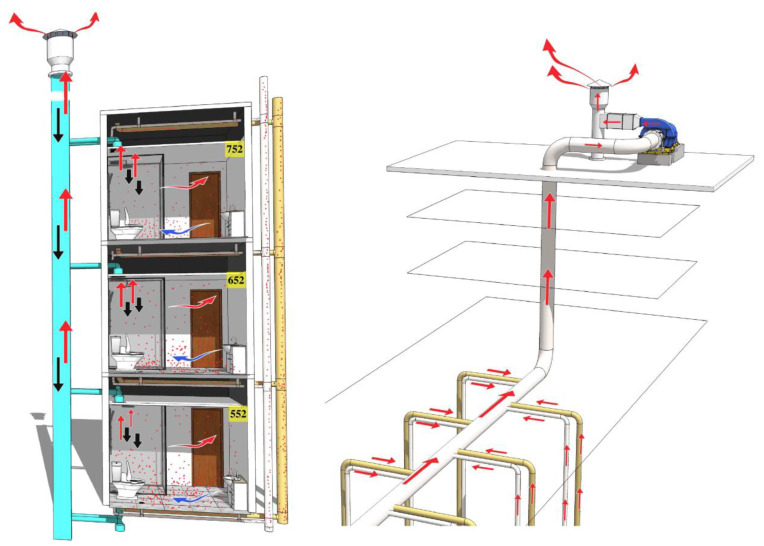
The schematic diagrams of vertical transmission in bathrooms and pipeline convergence. Yellow pipes indicate sewage pipes, white pipes indicate drainage pipes, blue pipes indicate exhaust pipes, and red dots indicate simulated viral aerosols.

**Figure 3 ijerph-19-01532-f003:**
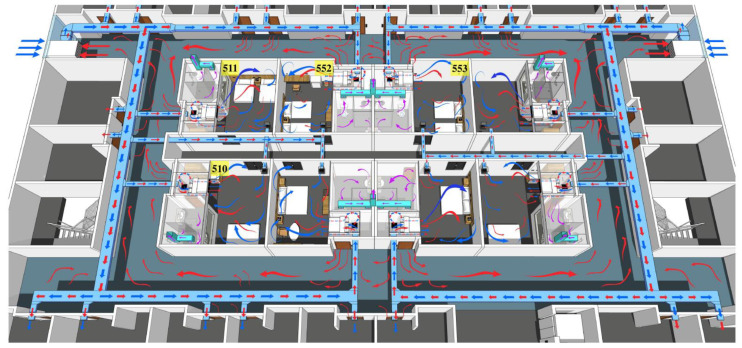
The layout of rooms in the hotel and schematic diagram of the transmission in the same layer. Blue arrows indicate fresh air from the outside, red arrows indicate the polluted air from rooms, and purple arrows indicate increasing contaminated air by toilet flushing.

**Table 1 ijerph-19-01532-t001:** The detection results of fluorescent microspheres of swab samples and air samples collected in different locations of different rooms in different time.

**Sample Location**	**Scenario 1/h**
**Room 552**	**Room 652**	**Room 752**
**0**	**0.5**	**1**	**1.5**	**0**	**0.5**	**1**	**1.5**	**0**	**0.5**	**1**	**1.5**
floor drain, toilet lid and seat ring												
Exhaust fan of bathroom												
air supply outlets of fresh air system and the air conditioning												
air sample (filter member)			
air sample (liquid)			
exhaust outlet of the roof	
sewage pipe riser vent on the roof	
**Sample Location**	**Scenario 2/h**
**Room 553**	**Room 652**	**Room 705**
**1**	**2**	**3**	**4**	**1**	**2**	**3**	**4**	**1**	**2**	**3**	**4**
air supply outlet of the air conditioning												
table, door, bed												
air supply outlets of fresh air system												
air sample (filter member)						
air sample (liquid)						
exhaust outlet of the roof	
sewage pipe riser vent on the roof	

## Data Availability

Not applicable.

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
