# Peer review of "Simulation Studies Provide Evidence of Aerosol Transmission of SARS-CoV-2 in a Multi-Story Building via Air Supply, Exhaust and Sanitary Pipelines"

_ijerph, 2022, doi:10.3390/ijerph19031532_

Round 1
Reviewer 1 Report
Did I get it right that some samples become negative again after having been positive (e.g. C4-C5, B5-B6)? Is this a sign that it is not entirely repeatable? Could you comment on this fact in the paper?
Fig. 1: Numbering of the room would be helpful in the figure.
Were the bioaerosol samplers and the PM10 samplers running thoughout the experiment? If so, a plot of their measurement might be helpful to underline the correlation with the activities you performed in the investigated room.
Is there an explanation why there is infections in 5th and 7th floor, but not in 6th? Can it be excluded that asymptomatic hotel staff brought the virus from 5th to 7th floor, especially because the ones on 5th floor had been there longer?
Supplementary material: What was your criterion for giving a + or a -? Which threshold did you apply? Is there a reason for not giving numeric values like concentration, number of particles, etc.?
I detected some small typos and commented into the pdf.

Author Response
Dear Reviewer,
Thank you for your comments and suggestions. Please see the attachment for the detailed response.

Reviewer 2 Report
Strength: Exceptionally good in terms of study conception, design, execution, and description.
Weakness: Needs more thorough justification and description description of the surrogate aerosol challenge, as the following:
Line Comment
29 replace “put” by “move out”.
131 specify the particle mean size (and size range) of the polystyrene microspheres. (see Line 265)
134 change “were” to “are”.
135 change “0h” to “0 h”.
149 change “adopted” to “in place”.
154 change “were” to “are”.
184 change “was” to “is”.
199 change “cannot” to “could not”.
210 change “can be” to “was”.
214 delete “will”, and change “pollute” to “polluted”.
220 replace “put” by “move out”.
226 replace “This result could be speculated” with “It can be speculated”.
230 change “was” to “is”.
247 change “2h” to “2 h”.
274 after “Nissen”, enter “et al. 2000” (22).
Author Response

(The authors gave the same response as above.)

Reviewer 3 Report
This manuscript describes an innovative study that measured a aerosol transmission of SARS-CoV-2 in a multi-story building 2 caused by air supply and exhaust and sanitary pipelines - simulation experiments provided sufficient evidences.
The objective of this study explores the cross-layer transmission path and influencing factors of viral aerosol, we set 27 up different scenarios to carry out simulation experiments.
This manuscript clearly falls within the scope of the International Journal of Environmental Research and Public Health.
Globally the paper is well written, the state-of-the-art is clearly presented and supported by references. But I think the authors can improve a little more the introduction.
The experimental section is presented clearly and in detail, but when referencing “agglomeration and resuspension of particles, while SARS-CoV-2 was a particle with a diameter of 100 nm (line 264), the humidity that can contribute to the aggregation or agglomeration of the ultrafine particles should be mentioned. The experimental studies could have evaluated nanoparticles with a Nanoparticle Surface Area Monitor (NSAM) or Nanometer Aerosol Sampler (NAS).
The results are well presented, which are widely discussed.
The conclusions could be further developed, there is a lot of interesting data in the article.
Even so, I recommend the publication of this manuscript in the International Journal of Environmental Research and Public Health.
Author Response

(The authors gave the same response as above.)
